# Quality of Life After Locoregional Treatment in Women with De Novo Metastatic Breast Cancer: A Systematic Review and Meta-Analysis

**DOI:** 10.3390/cancers17050751

**Published:** 2025-02-22

**Authors:** Camille Weiss, Philippe Trensz, Martin Schmitt, Massimo Lodi

**Affiliations:** 1Obstetrics and Gynecology Department, Strasbourg University Hospitals, 67200 Strasbourg, France; 2SOL Strasbourg Oncologie Liberale, 67000 Strasbourg, France; 3Radiation Therapy Department, Metz-Thionville Regional Hospital, 57530 Ars Laquenexy, France; 4Breast, Plastic and Reconstructive Surgery Department, Louis Pasteur Hospital, 68024 Colmar, France; 5Institut de Genetique et de Biologie Moleculaire et Cellulaire (IGBMC) Centre National de la Recherche Scientifique (CNRS UMR7104) Institut National de la Santé et de la Recherche Médicale (INSERM U964), Université de Strasbourg, 67400 Illkirch-Graffenstaden, France

**Keywords:** metastatic breast cancer, quality of life, surgery, primary tumor, meta-analysis

## Abstract

Breast cancer patients with metastatic disease may undergo surgery to remove the primary tumor, hoping to improve survival and quality of life. However, the actual benefits of this surgery remain uncertain. This study aimed to systematically review and analyze existing research to understand how surgical treatment of the primary tumor affects patients’ quality of life by examining several high-quality clinical trials. The findings challenge the common assumption that surgical intervention automatically improves patient well-being. This research is crucial because it provides important insights for doctors and patients when making treatment decisions, potentially helping to avoid unnecessary surgeries that might not provide the expected benefits and could even negatively impact a patient’s quality of life. These findings can be useful for patient counseling and multidisciplinary decision-making processes and can inform and prepare the patient” about potential quality of life impacts after treatment.

## 1. Introduction

The survival of patients with breast cancer depends on the stage of the disease: while 5-year survival is almost 96% for the early stages, it is only 38% for metastatic breast cancers (MBC, i.e., stage IV according to the Union for International Cancer Control classification) worldwide [1]. Even among MBCs, survival can vary and depend based on several factors, such as age, comorbidities, performance status, tumor molecular phenotype, extent and location of distant metastases, and treatments [2].

*De novo* MBC (dnMBC) accounts for approximately 3–6% [3,4] of newly diagnosed breast cancers and is different from subsequent MBC after initially localized breast cancer [3]. dnMBC refers to patients who are diagnosed with metastases at the time of their initial breast cancer diagnosis, often presenting with a better prognosis compared to those who develop metastases after treatment for an earlier, localized disease [3]. In contrast, subsequent MBC occurs in patients who have previously undergone locoregional treatment, such as surgery or radiotherapy, for non-metastatic breast cancer. In these cases, metastases typically arise without a concurrent local recurrence, meaning the primary tumor has already been managed. As a result, the role of additional locoregional treatment is generally less relevant in subsequent MBC.

The disease-specific survival of dnMBC has improved in recent decades [3]. Within the context of dnMBC, locoregional treatment at the primary site (LRT, including surgery +/− radiotherapy) has been performed and evaluated with the aim of improving survival, preventing complications, and alleviating local symptoms. Studies fail to show a consistent overall survival benefit in randomized controlled studies [5,6,7]. However, published data show an improvement in local management, with better local relapse-free survival [6,7]. Locoregional progression can be associated with pain, ulcerations, or lymphoedema, and one could expect that LRT should improve patients’ quality of life. However, LRT can also be responsible for adverse events and morbidity (such as breast cancer-related lymphoedema, pain, body image alteration, and psychological impact), which can affect the quality of life (QOL) negatively. In absence of local symptoms, this leads to the clinical dilemma that physicians and patients face: deciding between the potential QOL benefits of LRT due to improved local control and the risk of treatment-related side effects that could worsen the QOL. The clinical approach differs significantly when patients present with local symptoms such as skin invasion, ulceration, bleeding, pain, or lymphoedema. These symptoms and their management can substantially affect the quality of life (QOL) and may warrant surgical intervention for palliative purposes. In such cases, the decision to perform surgery is made on a case-by-case basis, aiming to alleviate symptoms and improve QOL.

Although the QOL is of paramount importance for medical interventions that fail to show a survival benefit, few studies have specifically investigated the QOL after non-palliative locoregional treatment at the primary site in patients with dnMBC, where surgery is considered in the absence of severe local symptoms for preventive purposes. Consequently, the aim of this study was to evaluate and quantify the impact of LRT on the QOL of patients with dnMBC through a systematic review of the literature and a meta-analysis. We wanted to conduct a comprehensive assessment of different dimensions of the QOL, such as physical, psychological, and social aspects, but also overall life satisfaction, and breast cancer- and treatment-specific factors.

## 2. Materials and Methods

We performed a systematic review of the literature and a meta-analysis according to the Preferred Reporting Items for Systematic reviews and Meta-Analyses Appendix A [8]. The methodology has been declared on the Open Science Framework Registry (https://osf.io/tz8m9, accessed on 21 February 2025).

### 2.1. Eligibility Criteria

Prospective, retrospective, and cohort studies were included if they met the following eligibility criteria:Population: patients with de novo metastatic breast cancer (treated with systemic therapy [ST]);Intervention: primary site LRT, defined as surgery with or without locoregional radiotherapy;Comparison: ST alone, i.e., the control group;Outcome: standardized assessment of QOL through any validated questionnaire.

In addition, to be included in the meta-analysis (quantitative synthesis), studies should provide sufficient quantitative data necessary for pooling results, specifically the means and standard deviations of QOL scores measured by validated instruments. Studies that lacked these data or used non-standardized measures were excluded from the pooled analysis. Finally, our study focuses exclusively on dnMBC, as unlike subsequent MBC it presents unique clinical considerations regarding the role of locoregional treatment as mentioned above.

### 2.2. Bibliographic Selection

The initial search was performed on various databases from inception until 10 May 2024 with the following keywords: (1) “metastatic breast cancer” or “stage IV breast cancer”; (2) “locoregional treatment” or “surgery” and (3) “quality of life”. This search yielded 265 articles on PubMed, 124 on the Cochrane Central Register of Controlled Trials, 107 on Web of Science, and 33 on Embase. We also reviewed the first 20 pages of Google Scholar queries (200 results) and the bibliographies of literature reviews published on the subject. After the removal of duplicates (*n =* 440), 289 articles were evaluated by 2 independent reviewers. There were no discrepancies between the two reviewers. Based on the title and abstract, 224 articles were excluded because they did not meet the inclusion criteria (see Figure 1). Therefore, we retained 17 articles for full-text reading. Of these 17 articles, 6 were excluded because they were systematic reviews of the literature or meta-analyses, 2 clinical trials had no results published at the time of the search, 1 article included only 2 patients, 1 had no control group, and 1 did not evaluate locoregional treatment. In total, 6 original articles were analyzed and 2 of them were excluded from the meta-analysis, as 1 did not directly compare treatments and the other did not provide enough QOL data for the analysis.

### 2.3. Data Collection and Statistical Analysis

For each article, we extracted the following information: author’s name and year of publication; number of patients included for QOL assessment; eligibility criteria; locoregional treatment modalities, including radical mastectomy rates, axillary lymph node dissection, breast reconstruction, and radiotherapy; and QOL measures and time of measure. For each article, we compared the QOL results between the LRT and control groups according to the time since inclusion. As QOL measures were not performed at the same time after inclusion across studies, we grouped timepoints into short-, intermediate- and long term (6, 18, and ≥30 months, respectively). The GRADE Grading of Recommendations Assessment, Development and Evaluation) methodology was used to assess the quality of evidence. The GRADE^®^ methodology systematically evaluates evidence certainty by initially rating randomized trials as high certainty and observational studies as low. It assesses five downgrading factors (risk of bias, inconsistency, indirectness, imprecision, publication bias) and considers upgrading factors for observational evidence (large effects, dose–response relationships). Final certainty levels range from high to very low.

Meta-analysis was performed using R (R version 4.2.1 (23 June 2022)) [9] with the metafor and metagear packages [10,11]. To compare QOL scores across studies, we used only validated QOL instruments (and for this reason, the study conducted by Si et al. [12] was included in the qualitative synthesis but excluded from the meta-analysis because it studied symptoms related to locoregional progression and not QOL directly). Despite variations between questionnaires, these scales assess similar core dimensions of QOL. Consequently, we compared the overall QOL assessment through the global score. In addition, the standardization of each score on a common metric was conducted prior to the analysis. Furthermore, given the heterogeneity of the populations and the different QOL scales used, a random-effects model was chosen for the meta-analysis.

As the included studies used different scales to measure QOL, we used the standardized mean difference (SMD) and its 95% confidence interval (CI) to estimate the magnitude of the effect and to reduce the risk of bias. The standard error was calculated using the formula *se = σ/√n*, where *σ* is the standard deviation and *n* is the number of subjects.

The studies were weighted according to the standard error of each population, which depended on the size and homogeneity. Heterogeneity was quantified using a maximum likelihood estimator for τ^2^, and we calculated the Higgins I^2^ statistic. For the heterogeneity test, the Cochran Q *p*-value was obtained with a Wald-type test. In one study, the standard deviation was missing and was imputed according to Bracken’s methodology [13], which estimates the standard deviation based on available data such as ranges or *p*-values. In another article [14], QOL data were described in two parts, on two scales: one covering physical aspects and another covering mental aspects. Both scales were comparable, as they were derived from the same questionnaire (SF-12). Therefore, to include both aspects, we calculated combined means and standard deviations for each group.

## 3. Results

In total, we included six studies for qualitative synthesis [12,15,16,17,18,19] (see Table 1 and Table 2, and Figure 2) and four of them also in quantitative analysis (meta-analysis, see Figure 3) [15,16,17,18].

Of interest, the included articles described surgical treatment characteristics (see Table 1). Breast-conserving surgery was performed in only 18–34% of cases, while axillary surgery was extensively performed (89–93%) as axillary lymph node dissection in the majority of cases (rates ranging from 65 to 93%).

### 3.1. Descriptive Analysis

The study by Khan et al. published in 2022 [16] (ECOG-ACRIN E2108) included 256 women with dnMBC treated with systemic treatment for 4–8 months, without progression between 2011 and 2015. The patients were then randomized into two groups: early local therapy (*n =* 125, although only 107 finally had LRT) *versus* continued ST (*n =* 131). In the LRT group, 29% of women had breast-conserving surgery, mostly with radiotherapy (87%); 71% had a total mastectomy; 91% underwent axillary surgery, including 76% with axillary lymph node dissection (ALND). In the continued ST group, 16.8% of the participants underwent surgery. The patient and tumor characteristics of the two groups were comparable, including direct invasion of the skin, skin nodules, and chest attachment. The quality of life was assessed using the TOI (Trial Outcome Index) of the Functional Assessment of Cancer Therapy FACT-B questionnaire, to which nine additional questions were added to cover arm symptoms and discomfort and worry related to locoregional disease. The authors found a significantly higher mean QOL score in the control group (continued ST) at 18 months compared to the LRT group (74.2 *versus* 68.0, *p* = 0.01); however, the difference was not observed at 6 months and at 30 months. Of interest, in the supplementary questions asked by the authors, there was a significant increase in arm stiffness and numbness in the LRT group.

The study by Bjelic-Radisic et al. published in 2020 [15] reports the QOL outcomes of the prospective randomized multicenter ABCSG-28 Posytive Trial [21]. It included 90 patients with operable dnMBC between 2011 and 2015, and 79 of them reported QOL through European Organisation for Research and Treatment of Cancer EORTC questionnaires (QLQ-C30 and QLQ-BR23) three times a year for 2 years; finally, 75 were included in the QOL analyses (ST alone, *n =* 41, *versus* LRT, *n =* 34). The authors found no significant differences between the two arms at any time point. LRT consisted of 29% of cases of breast-conserving surgery with radiotherapy, and 93% of the participants had axillary surgery, and among them 89% had ALND [21]. There were no statistically significant differences in the QOL scores between the groups at any time point. However, when the authors investigated the effect of time, they found worsening symptoms on the body image scale, clinically relevant in the surgery arm (*p* = 0.017).

In 2018, Soran et al. published a randomized controlled trial (protocol MF07-01) including 274 women with dnMBC treated with upfront LRT or ST only between 2007 and 2012 in Turkey [14]. LRT consisted of breast-conserving surgery and radiotherapy in 26% of the cases. All patients had axillary surgery, and among them, 92.8% had ALND. An ancillary article was published in 2021 [17] and analyzed QOL data 36 months after randomization in 81 patients (LRT, *n =* 55, and ST only, *n =* 26). The QOL was evaluated using the SF-12 questionnaire (divided into PCS-12 and MCS-122, respectively, Physical and Mental Health Composite Scales) and four additional questions on physical and mental health, daily activities, and energy. The authors found no significant differences between the two groups for the PCS-12 and MCS-12 scores (*p*-values were 0.34 and 0.54, respectively) and for the four additional questions (*p* range from 0.27 to 0.75).

In 2015, Badwe et al. published a randomized controlled trial [20] including 350 women with dnMBC aged ≤ 65 years old with an estimated remaining life expectancy of at least 1 year. In 2014, they published the results on the secondary outcome, which was the QOL [18]. The QOL was available for 178 women (84 in the surgery and 94 in the no-surgery groups). The groups were balanced and representative of patients in the main study. The authors found no significant difference between the groups at any time point (6 to 24 months) in global QOL. Interestingly, they found a significantly lower mean breast-specific QOL score in the surgery group at the 3–6 month evaluation, and this score was significantly higher during the 18–24 month evaluation.

In 2022, Chongxi et al. published [19] an abstract of a randomized controlled trial conducted in China on 81 women with dnMBC, evaluating initial surgery *versus* initial ST. Of interest, the primary endpoint of this study was the QOL, evaluated with the FACT-B TOI scale. The QOL was evaluated between 6- and 30-months post-randomization. They found that the average QOL scores showed no statistically significant difference between the two groups at each time point and concluded that LRT did not improve the QOL of this population.

Finally, in 2020, Si et al. published [12] a retrospective cohort study that included 177 individuals with dnMBC: *n =* 77 with upfront LRT and *n =* 100 without LRT. The primary endpoint was “local progression/recurrence symptoms”, defined as skin involvement (ulceration, bleeding, or discharge), and/or regional edema due to enlarged lesions or lymph node compression, and/or tumor-related moderate or severe pain in the breast or chest wall. The authors suggested that this measure was equivalent to the QOL and found that LRT significantly reduced the incidence of local symptoms (31.4 *versus* 68.6%, *p*-value 0.002). However, no other QOL evaluation was conducted in this cohort.

Table 2 describes the patient, tumor, and treatment characteristics of the studies included in the qualitative synthesis. Population characteristics and tumor histology were globally similar. The oligometastatic disease rate was heterogeneous (range 16–78%). Of interest, breast-conserving surgery rates were similar across studies and ranged from 18 to 34%. These rates are significantly lower compared to non-metastatic patients. Similarly, axillary surgery was almost always performed, and axillary lymph node dissection was the most frequent (65–93%). Systemic treatments were also different, with endocrine therapy ranging from 4 to 90% and chemotherapy from 31 to 96%.

### 3.2. Meta-Analysis

The joint analysis of global QOL in the four prospective studies [15,16,17,18] included 481 women with *n =* 251 in the LRT and *n =* 230 in the control groups (Figure 2 and Figure 3). Three studies investigated the short-term impact [15,16], and the pooled analysis did not find a significant QOL difference between groups (SMD = −0.51; 95%CI −1.55–0.53; *p* = 0.333, high heterogeneity I^2^ = 96%). Two studies also had available data at 18 months, and both studies found a statistically significant decrease in the QOL scores in the LRT group (SMD = −0.63; 95%CI −0.98–−0.26; *p* < 0.001, low heterogeneity I^2^ = 33%). Long-term data were available for four studies [15,16,17]. The pooled analysis showed a statistically significant decrease in the QOL of the LRT group (SMD −0.82; 95%CI −1.58–−0.06; *p* = 0.034, high heterogeneity I^2^ = 93%). Data from these articles show a numerically negative impact of LRT on the QOL, and this impact increases with time (see Figure 3). The pooled analysis was in favor of a consistent numerical decrease in the QOL with time in the LRT group compared to the control.

## 4. Discussion

This study shows that surgery for the primary tumor does not improve the quality of life of patients with *de novo* metastatic breast cancer. The results of this meta-analysis show a numerically lower QOL for patients that underwent LRT, although it is hard to estimate the clinical relevance of this result because the QOL measures are heterogeneous. Nevertheless, as the studies fail to show a survival benefit of LRT, the absence a QOL benefit questions the consequences of LRT for metastatic breast cancer patients. Therefore, this raises not only the issue of patient selection when considering LRT but also the modalities of LRT, especially regarding axillary dissection and the extent of radiation fields.

Recently, Ren et al. [22] published a meta-analysis that evaluated the impact of LRT on survival in dnMBC. They also included a meta-analysis of the QOL in two studies and found no difference between LRT and control groups. However, the authors did not include all randomized trials that reported QOL and did not evaluate the impact of LRT on the QOL over time.

### 4.1. Surgical Treatment Modalities

In managing MBC, systemic therapies—including endocrine chemo-, and targeted therapies—remain the primary treatment modality, with choices tailored to the tumor’s molecular subtype, due to their proven efficacy in controlling both local and distant disease spread [23]. The modalities of local treatment, especially surgery, are likely to affect the QOL, in particular if it does not prolong life expectancy. Most patients in the above-mentioned studies underwent mastectomy with ALND. No information was reported on immediate breast reconstruction in those trials, although it has been shown that breast reconstruction after mastectomy impacts short- and long-term QOL [24]. The literature shows that immediate breast reconstruction is less performed in MBC, although the rate is increasing. A Surveillance, Epidemiology, and End Results (SEER) study of 563 patients between 1998 and 2015 showed an increasing rate of immediate breast reconstruction of 6.3–16.8% and oncological safety, with no impact on overall and specific survival [25]. Regarding the reconstruction technique, another SEER study on 371 patients between 2004 and 2014 showed better specific and overall survival with implant-based reconstruction compared to autologous tissue [26], although this may be due to different patient characteristics. More recently, a predictive model also based on SEER was published [27]. Further studies are needed to investigate the prognostic impact of breast reconstruction. On the other hand, among the patients with breast-conserving therapy included in the selected articles, a clear margin rate between 76 and 100% was reported. There are very little published data on the impact of healthy margins’ benefit within this context [28]; therefore, additional surgical resection should be discussed in a multidisciplinary meeting if there are non-healthy margins for the infiltrating component.

Moreover, there are no recommendations regarding the management of the axilla for dnMBC patients. Data from patients with localized breast cancer show that ALND worsens the QOL compared to SLNB [29]. The prospective trials presented above include axillary surgery (including ALND) in more than 90% of cases. However, it has been demonstrated that ALND did not improve oncological outcomes in different randomized controlled trials in patients with stage I–III breast cancer [30,31], therefore suggesting that ALND has no intrinsic therapeutic benefit outside of staging and adapting therapeutic management. Axillary surgery (in particular ALND) has a significant impact on the QOL because of its well-known surgical morbidity (secondary lymphedema, pain, etc.), and this may explain the lower QOL rates in the LRT group. For these reasons, we support the fact that ALND can be avoided in dnMBC surgical management, even when the preoperative node status is positive (excepted when there is an axillary compression due to metastatic lymph nodes), as it does not impact therapeutic management and systemic therapy and may impair the QOL. Even if there are no randomized trials comparing ALND, SLNB and axillary surgical abstention in metastatic breast cancer, SLNB for patients with negative clinical and radiological nodal status may seem a reasonable option considering actual evidence for stage I-III breast cancer.

Finally, information on radiotherapy after surgery was not available in the included trials. Consequently, it is challenging to discern the individual contributions of surgery and radiotherapy to the observed results on the QOL. Given the evidence that adjuvant radiotherapy has minimal or no impact on the QOL [32,33] and the central role of systemic treatments in MBC, it is likely that results on the QOL across these studies are mainly explained by the surgical treatment.

### 4.2. Survival Benefit of LRT in Metastatic Disease

Although it has been hypothesized that removal of the primary tumor would lead to an improvement in overall survival, most recent studies and meta-analyses fail to show a consistent survival benefit. These studies were mainly retrospective and heterogeneous in terms of population, treatment, follow-up, and outcomes. In 2017, a Cochrane meta-analysis was published by Tosello et al. [7]. This meta-analysis included only two randomized controlled studies (624 patients) and found no difference in overall survival (hazard ratio [HR] 0.83; 95%CI 0.53–1.31), but a significant improvement in local progression-free survival (HR 0.22; 95%CI 0.08–0.57). On the contrary, distant progression-free survival was worse in the local treatment group (HR 1.42; 95%CI 1.08–1.86; one study), and the authors hypothesized that this was related to the interruption of systemic treatment. Then, another meta-analysis was published in 2020 by Gera et al. [5]. In this article, the authors included randomized controlled trials and retrospective studies. They reported an overall survival benefit of 36% (significant difference) when all studies were considered, but this benefit was 19% (non-significant) when only prospective studies were considered. The most recent meta-analysis published by Reinhorn et al. in 2021 [6], which included four randomized controlled trials (970 patients), found better local progression-free survival in the surgery group; however, it failed to demonstrate a benefit in overall survival, including after stratification on metastatic location and molecular phenotype.

Evidence of the benefit of locoregional treatment in metastatic breast cancer is difficult to establish, as not all MBCs have the same prognosis. Published studies have different objectives, and treatments (both local and systemic) are heterogeneous and in constant evolution. However, the absence of a demonstrated benefit in randomized trials should warn against this type of treatment outside of clinical trials. Nevertheless, LRT is regularly performed in dnMBC, and in up to 40% according to a French cohort study [34]. The European Society of Medical Oncology (ESMO) 2021 guidelines [35] state that locoregional treatment could be considered for asymptomatic women aged < 55 years with hormone receptor-positive HER2-negative bone oligometastatic breast cancer after a good response to initial systemic treatment. The pan-Asian ESMO 2023 guidelines [36] also state that local management could be considered in situations where it can treat a symptom or prevent a complication.

More recently, a novel approach was published by Bai et al. [37]. From a retrospective cohort of 7759 patients (from the SEER database and a Chinese database), the authors developed a predictive model based on a propensity score to estimate the survival benefit of surgery. This approach is promising; however, external and prospective validation is required before clinical applications. In view of the uncertain data on the overall survival benefit of locoregional treatment, the quality of life is an important factor. In addition, different surgical treatments and radiotherapy protocols exist, and the benefit of various modalities of treatment should be assessed when considering LRT.

Finally, another approach is the local treatment of oligometastatic sites, such as surgery and particularly stereotactic ablative radiotherapy (SABR). Recent evidence from randomized trials with a small number of metastatic breast cancer patients suggests a benefit of ablative therapies in terms of overall survival, and SABR is the therapy with the most supporting evidence [38]. In addition, this treatment has a good tolerance profile and is cost-effective [38]. This approach can also impact the QOL and is currently under investigation in the randomized controlled OLIGOMA trial [39].

### 4.3. Local Control Equals Better Quality of Life: An Incorrect Dogma

Does improvement in local control mean improvement in the QOL? This meta-analysis does not support this assertion, although local complications such as pain, ulceration, bleeding, or lymphoedema can be associated with local recurrence and likely worsen the QOL. Local recurrence can also be asymptomatic. For most patients, locoregional control of the primary tumor is achieved through systemic treatment [40].

The study by Si et al. [12] is interesting regarding this matter. The retrospective study concluded that there was an improvement in the QOL without proper QOL data. Indeed, the study’s primary endpoint was the occurrence of consequences (such as ulcerations, discharge, pain) due to local disease progression, and LRT showed improvement regarding the primary endpoint. The results of the retrospective study were affected by selection bias. The greater frequency of local complications in the arm without locoregional treatment seems to be linked to the operability of the tumor and its locoregional extension, and the two groups are not comparable in terms of the numbers of metastatic sites and lymph node extension.

The situation is different when local symptoms—such as skin invasion, ulceration, bleeding, pain, lymphoedema—are already present. These local symptoms and their treatments negatively impact the QOL and may be addressed through surgical treatment. Within this context of palliative surgery, the indication for surgery is decided on a case-specific basis and may improve the QOL [40]. In addition, it should be emphasized that the QOL data presented in this article relate to the indication for non-palliative surgery and therefore cannot be applied in the decision for palliative local surgery, which is generally guided by disease progression and local symptoms.

This shows that when discussing LRT, we focus on a local recurrence benefit or a survival benefit, but the significant impact of LRT on the QOL must also be taken into account. For instance, we know that ALND is associated with consequences that have been linked to a worsening QOL [29]. Data from this meta-analysis show that while LRT does indeed appear to reduce locoregional disease progression and, thus, its consequences, it also appears to be a source of impaired QOL.

Therefore, the choice of performing LRT in dnMBC should be nuanced on a case-by-case basis, balancing between potential benefits and harm to both survival and the QOL. In cases of local symptoms which impair the QOL, surgical control may offer—sometimes transient—relief and improve the QOL. On the contrary, curative-like LRT in the absence of local symptoms raises concerns about added surgical morbidity without significant survival benefits in all situations. Hence, treatment strategies must prioritize patient-centric outcomes, such as the quality of life (QOL), and consider the individual’s overall health status, disease progression, and personal preferences. This underscores the importance of multidisciplinary decision making tailored to each patient’s unique clinical context.

### 4.4. Strengths and Limitations

The studies included in the meta-analysis are heterogeneous, both in terms of the local treatment received and in terms of the QOL assessment. Indeed, the QOL was measured with different scales and it was therefore harder to compare it between studies. Also, the completion rate of the QOL questionnaires undermined the validity of the results. At 18 months, in the studies by Kahn et al. [16] and Bjelic-Radisic et al. [15], only 50% and 56% of patients, respectively, had QOL data available. The difficulty of obtaining quality data over an extended period of time in a patient population is a frequent problem [41], which becomes more acute as one moves away from the baseline value. It is likely that a low completion rate leads to a selection bias that affects the validity of the results [41]. However, it could be assumed that local complications are responsible for a possible deterioration in the QOL that occurs late in the course of metastatic breast cancer, when systemic treatment is no longer able to control the disease. Current data do not support this assumption, since data beyond 18 months are insufficient and the median life expectancy across all histological types is 37 months [42]. The included studies used different validated QOL instruments, and although some items overlap, there were differences within the proposals, making the analysis less homogeneous. Each instrument varies in its focus, scales, and sensitivity to changes in QOL dimensions. This variability can influence the results, as some instruments may capture certain aspects of the QOL more effectively than others. The SF-12 questionnaire, for example, contains fewer questions than the other two questionnaires, making it less precise. However, all questionnaires were validated to measure the QOL, and as there is no official gold standard, the choice of a suitable questionnaire remains open. This point is also reflected in the high heterogeneity observed in the meta-analysis.

Metastatic breast cancer concerns a heterogeneous population of patients in terms of demographic, tumoral, and treatment characteristics. The studies included in this meta-analysis did not report the rate of oligometastatic diseases at baseline. We also do not know how many patients were in complete iconographic response at the time of LRT. We also do not know how many patients were in complete histologic response at the time of surgery. We believe that this is likely to influence the results of our study. What are the QOL benefits of surgery when systemic therapy has been efficient enough to allow for complete histologic response? Whether or not LRT is associated with a survival benefit in patients without evidence of metastatic disease is a matter of debate, which reinforces the need for extensive QOL data. Finally, the included studies varied in surgical practices, with differences in the rates of mastectomy *versus* breast-conserving surgery, and variance in the extent of axillary surgery and immediate breast reconstruction. There are no reported data regarding the volumes and doses of radiation therapy. All these factors have an impact on the quality of life [43,44,45], and these variations can introduce biases.

## 5. Conclusions

This study shows that there is lacking evidence regarding the QOL benefits after LRT for *de novo* metastatic breast cancer patients. The results even show a numerical deterioration in the global QOL in the locoregional treatment group several months after the treatment. Even though it is hard to estimate the clinical relevance of this QOL reduction with time, it is unlikely that a better local control is not necessarily associated with better QOL. These findings can be useful for patient counseling and multidisciplinary decision-making processes. Given that treatment for patients with metastatic breast cancer is chronic, these results can assist in deciding if surgery has to be performed and also in planning the timing. They can also inform and prepare the patient for this negative impact on QOL afterward. Moreover, these findings challenge a common belief that LRT inherently improves QOL in metastatic breast cancer patients.

Further randomized studies comparing surgery to no surgery are currently underway, and QOL data may help to evaluate the benefit of LRT for patients suffering from a metastatic disease. [46,47,48,49]. In addition, future research should emphasize identifying patient subgroups that might benefit most from LRT, considering both the QOL and survival outcomes, especially in long-term survivors. Understanding these nuances can aid in developing more personalized treatment approaches and improving the overall management of metastatic breast cancer.

## Figures and Tables

**Figure 1 cancers-17-00751-f001:**
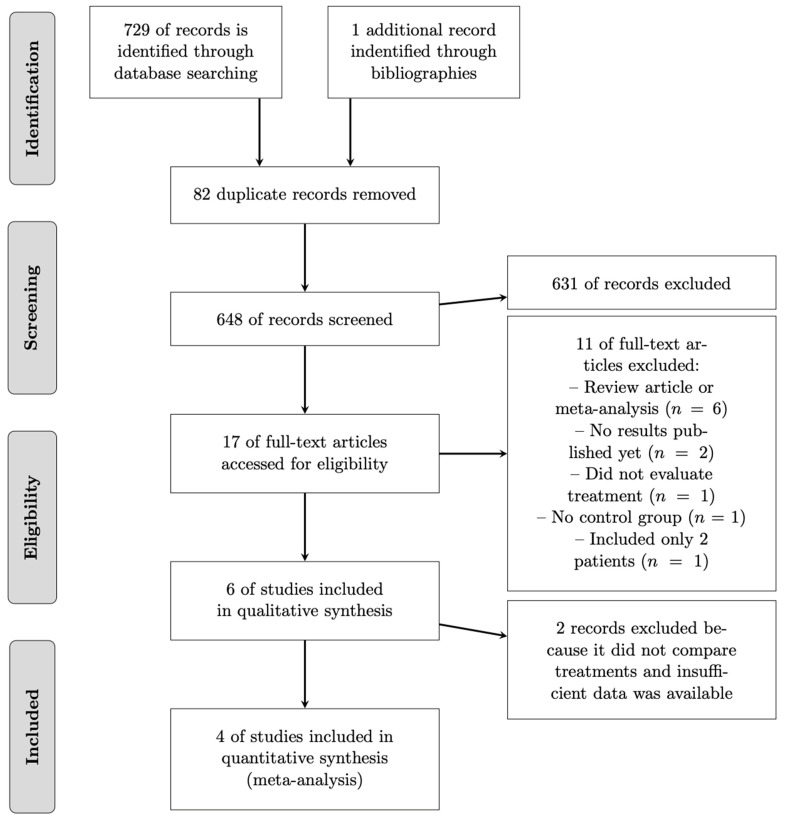
Flowchart.

**Figure 2 cancers-17-00751-f002:**
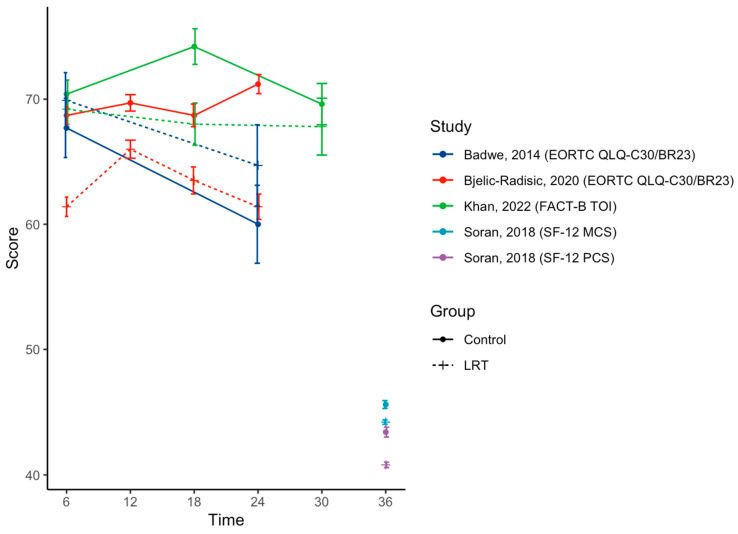
Quality of life according to time (months) since baseline. Legend: LRT = locoregional treatment [14,15,16,20].

**Figure 3 cancers-17-00751-f003:**
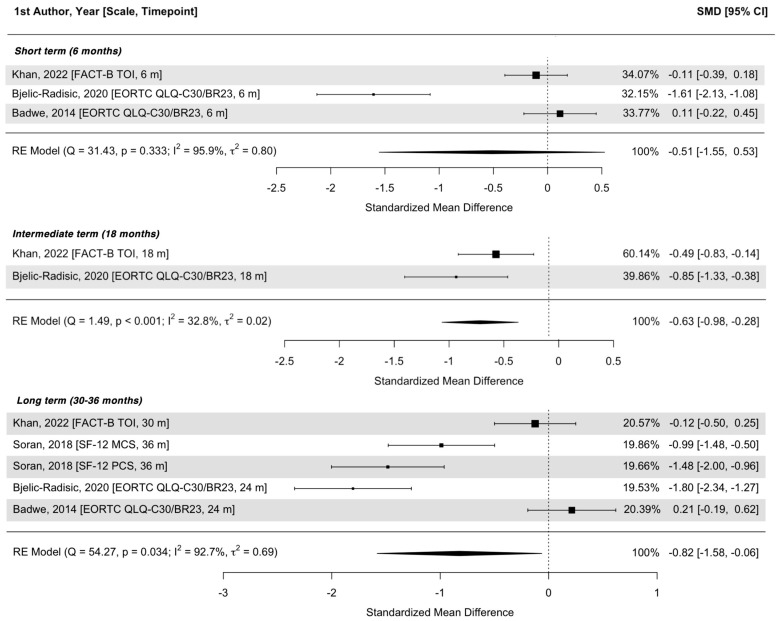
Quality of life differences between the locoregional treatment and the control groups according to time since baseline. Legend: SMD = standardized mean difference; RE = random effect; m = months [14,15,16,20].

**Table 1 cancers-17-00751-t001:** Summary of included studies with GRADE rating of the certainty of evidence.

Author, Year	Study Design	Certainty Assessment	Summary of Findings on QOL	Certainty
Khan, 2022 [16]	RCT (follow-up: mean 53 months); Years: 2011–2015; Country: United States; Inclusion: dnMBC with systemic therapy for 4 to 8 months without progression; Total patients: 256 (Early local therapy *n =* 125; *versus* Continued ST *n =* 131); Primary outcome: overall survival; QOL scale: FACT-B TOI [range 0–96].	Risk of bias: not serious; Inconsistency: not serious; Indirectness: not serious; Imprecision: not serious; Other considerations: none	QOL was significantly higher at 18 months post-randomization in the Continued ST group, without any significant difference at other time points.	High
Bjelic-Radisic, 2020 [15]	RCT (follow-up: mean 37.5 months); Years: 2011–2015; Country: Austria; Inclusion: dnMBC; Total patients: 79 (upfront LRT *n =* 37; *versus* initial ST *n =* 42); Primary outcome: overall survival; QOL scale: EORTC QLQ-C30 and BR23 [range 0–100].	Risk of bias: serious ^a,b^; Inconsistency: not serious; Indirectness: not serious; Imprecision: not serious; Other considerations: none	There were no statistically significant differences in QOL between the two groups over time.	Moderate
Soran, 2021 [17]	RCT (follow-up: mean 40 months); Years: 2007–2012; Country: Turkey; Inclusion: dnMBC living ≥36 months after randomization; Total patients: 274 (upfront LRT *n =* 55; *versus* ST only *n =* 26); Outcome: QOL; QOL scale: MCS-12 and PCS-12 [range 0–100].	Risk of bias: serious ^c^; Inconsistency: not serious; Indirectness: not serious; Imprecision: not serious; Other considerations: none	No significant difference between groups.	Moderate
Badwe, 2014 [18]	RCT (follow-up: mean 23 months); Years: 2005–2013; Country: India; Inclusion: dnMBC aged ≤ 65 years months; Total patients: 178 (upfront surgery *n =* 84; *versus* ST only *n =* 94); Outcome: QOL; QOL scale: EORTC QLQ-C30 and BR23 [range 0–100].	Risk of bias: not serious; Inconsistency: not serious; Indirectness: not serious; Imprecision: not serious; Other considerations: abstract only ^d^	No significant difference between groups at any time point (6 to 24 months) in global QOL. Lower breast-specific QOL in the surgery group at 3–6 month evaluation, and higher at 18–24 month evaluation.	Moderate ^d^
Chongxi, 2022 [19]	RCT (follow-up: mean 37.5 months); Years: 2019–2021; Country: China; Inclusion: dnMBC; Total patients: 81 (initial surgery *n =* 41; *versus* initial ST *n =* 40); Primary outcome: quality of life; QOL scale: FACT-B TOI	Risk of bias: not serious; Inconsistency: not serious; Indirectness: not serious; Imprecision: not serious; Other considerations: abstract only ^d^	No significant difference between groups at any time point (6 to 30 months).	Moderate ^d^
Si, 2020 [12]	Cohort, retrospective (follow-up: median 33 months); Years: 2008–2014; Country: China; Inclusion: dnMBC; Total patients: 193 (LRT *n =* 77; *versus* ST only *n =* 100); Outcome: local progress/recurrence of symptoms	Risk of bias: serious ^b,e^; Inconsistency: not serious; Indirectness: serious ^f^; Imprecision: serious ^f^; Other considerations: none	Surgery reduced progress/recurrence of symptoms (RR = 0.440; *p* = 0.015), a surrogate measure of QOL.	Low

^a^ The study stopped prematurely at 4 years because of slow recruitment; ^b^ low number of patients; ^c^ selection bias of only patients living ≥36 months after randomization; ^d^ as only the abstract was available at the time of this meta-analysis, a complete certainty assessment was not possible; therefore, we downgraded the quality of evidence to moderate instead of high; ^e^ retrospective; ^f^ indirect measure of QOL. Legend: ST = systemic therapy; RCT = randomized controlled trial; dnMBC = de novo metastatic breast cancer; QOL = quality of life; RR = relative risk; EORTC = European Organisation for Research and Treatment of Cancer; FACT = Functional Assessment of Cancer Therapy; PCS, MCS = physical, mental component summary of the Short Form 36 (SF-36) Health Survey.

**Table 2 cancers-17-00751-t002:** Patient, tumor, and treatment characteristics.

Author, Year	Demographics		Tumor Characteristics					Treatment Characteristics				
	Mean Age (Years)	Postmenopausal Status (Rate)	Luminal	Triple Negative	HER2-Positive	Oligometastatic Disease	Locally Advanced Tumor	Endocrine Therapy	Chemotherapy	BCS	Axillary surgery	ALND
Khan, 2022 [16]	56	64%	60%	8%	32%	16%	48%	41%	68%	18%	LRT = 93% vs. Control = 16%	65%
Bjelic-Radisic, 2020 [15]	62	87%	71%	10%	19%	--	35%	68%	31%	29%	93%	89%
Soran, 2021 [17]	54	--	--	--	--	--	--	90%	95%	26%	100%	93%
Badwe, 2014 [18]	48	53%	--	--	30%	25%	--	4%	96%	--	--	--
Chongxi, 2022 [19]	--	--	--	--	--	--	--	--	--	--	--	--
Si, 2020 [12]	--	48%	--	--	49%	78%	46%	--	--	34%	--	--

LRT = locoregional treatment; BCS = breast-conserving surgery; ALND = axillary lymph node dissection.

## Data Availability

The data that support the findings of this study are derived from previously published studies, which are all referenced in the manuscript. These prior studies are openly available in various public databases. As this is a meta-analysis, no new original data were created in this study. Collected data and R code are available upon request.

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
