# Peer review of "Quality of Life After Locoregional Treatment in Women with De Novo Metastatic Breast Cancer: A Systematic Review and Meta-Analysis"

_cancers, 2025, doi:10.3390/cancers17050751_

Round 1

Reviewer 1 Report

Comments and Suggestions for Authors

General Comments:
This topic addresses an important area by analyzing past research on the quality of life of patients with de novo metastatic breast cancer (MBC). However, similar reviews with comparable conclusions have already been published, such as those by Boaitey et al. (2024), Ren et al. (2024), and Reinhorn et al. (2021) [[1][2][3]]. This manuscript offers minimal new insights, if any, to the field.

Specific Comments:

  1. The manuscript is riddled with instances of “Error! Reference source not found.” The authors should thoroughly review the manuscript to resolve these issues before submission. The missing references significantly hinder readability and create confusion for reviewers.
  2. The manuscript highlights that de novo MBC differs from MBC that arises after initially localized breast cancer. However, it fails to justify why the focus is solely on de novo MBC and excludes subsequent MBC. The authors should provide a clear clinical rationale for this decision and its significance.
  3. In many of the included studies, systemic therapy (ST) is used as a control group, while interventions involve various surgical procedures. The manuscript would benefit from a more detailed discussion of the clinical rationale behind the surgical interventions and why ST alone serves as the control. Additionally, the inclusion of randomized controlled trials (RCTs) involving surgeries is puzzling, given that surgical decisions are typically guided by disease progression rather than random allocation. This requires clarification.
  4. The numbers in Figure 1 are inconsistent and lack adequate explanation. For instance, after “289 records after duplicates removed,” the flowchart inexplicably proceeds to “648 records screened.” The discrepancy between these numbers needs to be clarified.
  5. Numerous acronyms, such as EORTC, ALND, and FACT-B, are not explained in the text. All acronyms should be defined upon first use to ensure clarity for readers unfamiliar with these terms.

[1] Boaitey, G.A., Martini, R., Stonaker, B. et al. Patterns of breast cancer locoregional relapse, metastasis, and subtypes in Ghana. BMC Cancer 24, 1485 (2024). https://doi.org/10.1186/s12885-024-13254-x

[2] Chongxi Ren, Jianna Sun, Lingjun Kong, Hongqiao Wang,

Breast surgery for patients with de novo metastatic breast cancer: A meta-analysis of randomized controlled trials, European Journal of Surgical Oncology, Volume 50, Issue 1, 2024, 107308, ISSN 0748-7983,

[3] Reinhorn D, Mutai R, Yerushalmi R, Moore A, Amir E, Goldvaser H. Locoregional therapy in de novo metastatic breast cancer: Systemic review and meta-analysis. Breast. 2021 Aug;58:173-181.

Author Response

We sincerely appreciate the time and expertise invested in reviewing our work. Please find below the comprehensive responses to each query raised, accompanied by highlighted revisions in the manuscript files to reflect adjustments made in light of your recommendations.

Comment 1: This topic addresses an important area by analyzing past research on the quality of life of patients with de novo metastatic breast cancer (MBC). However, similar reviews with comparable conclusions have already been published, such as those by Boaitey et al. (2024), Ren et al. (2024), and Reinhorn et al. (2021) [[1][2][3]]. This manuscript offers minimal new insights, if any, to the field.

Response 1: we thank the reviewer for the reviewing work and thoughtful comments, which have improved the quality of this article. We appreciate the opportunity to address the concerns raised regarding the novelty of our study. Upon careful consideration of the reviewer's comments, we would like to respectfully clarify the distinctions between our work and the cited references:

  1. Reinhorn et al. (2021) investigated the overall survival and not the quality of life, which is the central theme of our study.
  2. While Ren et al. (2024) did publish data on quality of life, however they did not include all randomized trials that reported QOL and did not evaluate the impact of LRT on QOL over time, as stated in the discussion of our study.
  3. Regarding Boaitey et al. (2024), we found that this reference is not directly relevant to our objective. It appears to be a retrospective study on patterns of locoregional and distant relapse, which falls outside the scope of our research.

Comment 2: The manuscript is riddled with instances of “Error! Reference source not found.” The authors should thoroughly review the manuscript to resolve these issues before submission. The missing references significantly hinder readability and create confusion for reviewers.

Response 2: we apologize for the inconvenience as this error did not show in our version of the manuscript, we removed the references for the tables and figures and replaced it with plain text

Comment 3: The manuscript highlights that de novo MBC differs from MBC that arises after initially localized breast cancer. However, it fails to justify why the focus is solely on de novo MBC and excludes subsequent MBC. The authors should provide a clear clinical rationale for this decision and its significance.

Response 3: Subsequent MBC is defined as the occurrence of distant metastases after treated non-metastatic primary breast cancer. Given that surgery is an indispensable step in the treatment of non-metastatic breast cancer, patients with subsequent MBC have necessarily already undergone locoregional treatment, including surgery ± radiotherapy. We excluded these patients because a history of breast surgery would have been a major bias in assessing the quality of life of a new locoregional treatment, as well as the fact that most often there is not necessarily a new local recurrence and therefore the question of locoregional treatment does not arise. We added a sentence in the methodology to avoid any confusion.

Comment 4: In many of the included studies, systemic therapy (ST) is used as a control group, while interventions involve various surgical procedures. The manuscript would benefit from a more detailed discussion of the clinical rationale behind the surgical interventions and why ST alone serves as the control. Additionally, the inclusion of randomized controlled trials (RCTs) involving surgeries is puzzling, given that surgical decisions are typically guided by disease progression rather than random allocation. This requires clarification.

Response 4: we thank the reviewer for this insightful comment, and we agree with it. The confusion arises from the fact that in the discussion there was no distinction between curative-like surgery (i.e. the scope of this article) and palliative surgery. Indeed, palliative surgery is indicated for local symptoms and progression. However, surgery is also performed at the beginning of the treatment with the aim to improve survival or to prevent local symptoms. In this situation, RCT are an efficient way to evaluate surgery, but could not have been applied for palliative surgery contexts. We therefore made a clarification in the discussion: “ The situation is different when local symptoms – such as skin invasion, ulceration, bleeding, pain, lymphoedema – are already present. Those local symptoms and their treatments negatively impact QOL, and may be accessible to surgical treatment. Within this context of palliative surgery, the indication for surgery is decided on a case-specific basis and may improve QOL (39). In addition, it should be emphasized that the QOL data presented in this article relate to the indication for non-palliative surgery, and therefore cannot be applied in the decision for palliative local surgery, which is generally guided by disease progression and local symptoms.”

Comment 5: The numbers in Figure 1 are inconsistent and lack adequate explanation. For instance, after “289 records after duplicates removed,” the flowchart inexplicably proceeds to “648 records screened.” The discrepancy between these numbers needs to be clarified.

Response 5: we thank the reviewer for pointing this error out, we corrected the flowchart accordingly

Comment 6: Numerous acronyms, such as EORTC, ALND, and FACT-B, are not explained in the text. All acronyms should be defined upon first use to ensure clarity for readers unfamiliar with these terms.

Response 6: we agree with the reviewer and completely reviewed the manuscript, figures and tables, and added the explanations for acronyms

Reviewer 2 Report

Comments and Suggestions for Authors

The authors demonstrated the QOL following LRT in dnMBC, and the data appeared to be reliable.

Overall, LRT did not provide a benefit to QOL. However, I believe that LRT could offer benefits to patients with breast cancer involving skin invasion, ulceration, or bleeding. Therefore, I recommend that the authors discuss the effects of LRT on such patients in the Discussion section.

Author Response

Comment 1: The authors demonstrated the QOL following LRT in dnMBC, and the data appeared to be reliable. Overall, LRT did not provide a benefit to QOL. However, I believe that LRT could offer benefits to patients with breast cancer involving skin invasion, ulceration, or bleeding. Therefore, I recommend that the authors discuss the effects of LRT on such patients in the Discussion section.

Response 1: we thank the reviewer for this insightful comment and we agree with, this was a gap in the discussion of the article and we made a clarification in the discussion (4.3 subsection, paragraph 3): “ The situation is different when local symptoms – such as skin invasion, ulceration, bleeding, pain, lymphoedema – are already present. Those local symptoms and their treatments negatively impact QOL, and may be accessible to surgical treatment. Within this context of palliative surgery, the indication for surgery is decided on a case-specific basis and may improve QOL (39). In addition, it should be emphasized that the QOL data presented in this article relate to the indication for non-palliative surgery, and therefore cannot be applied in the decision for palliative local surgery, which is generally guided by disease progression and local symptoms.”

Reviewer 3 Report

Comments and Suggestions for Authors

Comments regarding the systematic review titled: Quality of life after locoregional treatment in women with de novo metastatic breast cancer: a systematic review and meta-analysis

The authors present a systematic review focusing in de novo metastatic breast cancer patients’ quality of life after undergoing removal of the primary tumor. The main purpose of the authors is to understand how surgical treatment of the primary tumor affects patients' quality of life. The review is very relevant in field of metastatic breast cancer and has the potential to impact treatment decisions regarding surgery in this patient population.

Some comment:

Section 2.2: Is there a time frame from which the studies were selected, for example last 5 years?

Figure 1: Revise the numbers in the flow chart. If 648 records were screened and 638 excluded, only 10 records remained. Also, in the text it states that there were 17 full-text articles but in the flowchart, it appears as 16.

Results section: Figures and tables are references within the text. Please verify.

Table 1 mentions a Certainty assessment. The criteria to perform this assessment need to be included in the Methods section. Also, describe how do you categorize certainty as high or moderate. It will improve the reader’s understanding of the table.

Please check the references across the manuscript. Some information is missing, and error messages appear within the text.

Author Response

We sincerely appreciate your thorough review of our manuscript. Below, you'll find our comprehensive responses to your comments, along with the corresponding revisions clearly marked in the resubmitted documents.

Comment 1: Section 2.2: Is there a time frame from which the studies were selected, for example last 5 years?

Response 1: we agree with the reviewer as this information was missing, we corrected the methodology section as there was no time frame for study selection.

Comment 2: Figure 1: Revise the numbers in the flow chart. If 648 records were screened and 638 excluded, only 10 records remained. Also, in the text it states that there were 17 full-text articles but in the flowchart, it appears as 16.

Response 2: we thank the reviewer for pointing this error out, we corrected the flowchart accordingly

Comment 3 and 5: Results section: Figures and tables are references within the text. Please verify. Please check the references across the manuscript. Some information is missing, and error messages appear within the text

Response 3 and 5: we apologize for the inconvenience as this error did not show in our version of the manuscript, we removed the figures/tables references for the tables and figures and replaced it with plain text. 

Comment 4: Table 1 mentions a Certainty assessment. The criteria to perform this assessment need to be included in the Methods section. Also, describe how do you categorize certainty as high or moderate. It will improve the reader’s understanding of the table.

Response 4: we agree with the reviewer and provided an explanation in the methodology (subsection 2.3, paragraph 1): “The GRADE® methodology systematically evaluates evidence certainty by initially rating randomized trials as high certainty and observational studies as low. It assesses five downgrading factors (risk of bias, inconsistency, indirectness, imprecision, publication bias) and considers upgrading factors for observational evidence (large effects, dose-response relationships). Final certainty levels range from high to very low.”

Round 2

Reviewer 1 Report

Comments and Suggestions for Authors

The authors have made a commendable effort in addressing the previous reviewer’s comments. However, many of the revisions have not been appropriately incorporated into the manuscript. For instance, the response to comment 4 does not appear to be reflected in the revised version.

In the Discussion section, the authors added a paragraph explaining the consideration of treatment regarding local symptoms. However, additional background information should be provided in the Introduction to clarify the clinical considerations of de novo metastatic breast cancer (dnMBC) compared to other types of breast cancer. Specifically, it is important to address whether surgery is a necessary treatment from a clinical perspective.

Furthermore, quality of life (QOL) is closely associated with overall health status and life expectancy following diagnosis. If surgery does not prolong life expectancy, it is evident that it would only add to the patient’s burden. Therefore, incorporating a discussion on the clinical rationale behind different treatment strategies for various types of breast cancer would enhance the manuscript’s readability and accessibility for a broader audience.

Lastly, in Figure 1, the text in the box "82 of records after duplicates removed" is unclear. It should be revised to "82 duplicate records removed" for better clarity.

Comments on the Quality of English Language

English language is fine in general.

Author Response

We thank again the reviewer’s careful reading and comments, and its constructive feedback. Below, we present our detailed responses to each comment, with all corresponding revisions clearly marked in the manuscript and reported at the end of this response.

Comment 1: The authors have made a commendable effort in addressing the previous reviewer’s comments. However, many of the revisions have not been appropriately incorporated into the manuscript. For instance, the response to comment 4 does not appear to be reflected in the revised version.

Response 1: we agree with the reviewer and upon review, we have identified that the response to previous (R1) comments 3 and 4 were indeed not fully integrated into the manuscript. We have now thoroughly incorporated the suggested changes and moved them in the introduction section, ensuring that these points are clearly reflected. Regarding comment 3, we detailed the rationale for focusing only on dnMBC instead of subsequent MBC in the introduction 1 and we kept a phrase in the methodology 2. Regarding comment 4, we moved the paragraph from the discussion to the end of the introduction, to highlight the rationale of evaluation preventive/curative-like/systematic locoregional treatment 3,4.

Comment 2: In the discussion section, the authors added a paragraph explaining the consideration of treatment regarding local symptoms. However, additional background information should be provided in the Introduction to clarify the clinical considerations of de novo metastatic breast cancer (dnMBC) compared to other types of breast cancer. Specifically, it is important to address whether surgery is a necessary treatment from a clinical perspective.

Response 2: we thank the reviewer for this suggestion, and we agree with it. Therefore, as in the comment before, we added several paragraphs in the introduction from the discussion and the methodology as it was part of the clinical rationale and the background. This addition provides a clearer context for understanding the rationale behind different treatment strategies.

Comment 3: Furthermore, quality of life (QOL) is closely associated with overall health status and life expectancy following diagnosis. If surgery does not prolong life expectancy, it is evident that it would only add to the patient’s burden. Therefore, incorporating a discussion on the clinical rationale behind different treatment strategies for various types of breast cancer would enhance the manuscript’s readability and accessibility for a broader audience.

Response 3: we agree with the reviewer and therefore we revised the discussion section to include a more detailed  explanation of the clinical rationale behind different treatment strategies, emphasizing the role of QOL and its relationship to life expectancy 5,6. These addition highlights how decisions regarding surgery in dnMBC are made with benefit/risk balance on a case-by-case basis, thereby making it more accessible to broader audience.

Comment 4: Lastly, in Figure 1, the text in the box "82 of records after duplicates removed" is unclear. It should be revised to "82 duplicate records removed" for better clarity.

Response 4: we thank the reviewer for pointing this out. We have corrected the text in Figure 1 from "82 of records after duplicates removed"to "82 duplicate records removed" to improve clarity

Modifications made in the manuscript regarding comments 1 and 2:

  • 1 “dnMBC refers to patients who are diagnosed with metastases at the time of their initial breast cancer diagnosis, often presenting with a better prognosis compared to those who develop metastases after treatment for an earlier, localized disease (3). In contrast, sub-sequent MBC occurs in patients who have previously undergone locoregional treatment, such as surgery or radiotherapy, for non-metastatic breast cancer. In these cases, metastases typically arise without a concurrent local recurrence, meaning the primary tumor has already been managed. As a result, the role of additional locoregional treatment is generally less relevant in subsequent MBC.”
  • 2 “Finally, our study focuses exclusively on dnMBC, as unlike subsequent MBC it presents unique clinical considerations regarding the role of locoregional treatment as mentioned above.”
  • 3 “The clinical approach differs significantly when patients present with local symptoms such as skin invasion, ulceration, bleeding, pain, or lymphoedema. These symptoms and their management can substantially affect quality of life (QOL) and may warrant surgical intervention for palliative purposes. In such cases, the decision to perform surgery is made on a case-by-case basis, aiming to alleviate symptoms and improve QOL. Although QOL is of paramount importance for medical interventions that fail to show a survival benefit, few studies have specifically investigated QOL after non-palliative locoregional treatment at the primary site in patients with dnMBC, where surgery is considered in the absence of severe local symptoms for preventive purposes.”
  • 4 “Although QOL is of paramount importance for medical interventions that fail to show a survival benefit, few studies have specifically investigated QOL after non-palliative locoregional treatment at the primary site in patients with dnMBC, where surgery is considered in the absence of severe local symptoms for preventive purposes.”

Modifications made in the manuscript for comment 3:

  • 5 at the beginning of the subsection 4.1: “In managing MBC, systemic therapy – including endocrine chemo-, and targeted therapies – remain the primary treatment modality, with choices tailored to the tumor's molecular subtype, due to their proven efficacy in controlling both local and distant disease spread (23). Modalities of local treatment, especially surgery, are likely to affect QOL in particular if it does not prolong life expectancy.”
  • 6 at the end of the subsection 4.3: “Therefore, the choice of performing LRT in dnMBC should be nuanced on a case-by-case basis, balancing between potential benefits and harms both on survival and QOL. In cases of local symptoms which impair the QOL, surgical control may offer – sometimes transient – relief, and improve QOL. On the contrary, curative-like LRT in the absence of local symptoms raises concerns about added surgical morbidity without significant survival benefits in all situations. Hence, treatment strategies must prioritize patient-centric outcomes, such as quality of life (QOL), and consider the individual's overall health status, disease progression, and personal preferences. This underscores the importance of multidisciplinary decision-making tailored to each patient's unique clinical context.”